# Nationwide Long-Term Evaluation of Polypharmacy Reduction Policies Focusing on Older Adults in Japan

**DOI:** 10.3390/ijerph192214684

**Published:** 2022-11-09

**Authors:** Takehiro Ishida, Asuka Suzuki, Yoshinori Nakata

**Affiliations:** Graduate School of Public Health, Teikyo University, Itabashi-ku, Tokyo 173-8605, Japan

**Keywords:** aged, appropriate prescribing, health policy, polypharmacy, prescription fees, public health, real-world evidence

## Abstract

Polypharmacy is a serious health issue for older adults worldwide, including in Japan, which has a rapidly aging society. The “Proper Medication Guideline for Older Adults” was published for healthcare providers in May 2018, and polypharmacy reduction incentives were initiated for medical facilities in April 2016 and April 2018. This study identified the long-term reduction in polypharmacy prescriptions focusing on older adults aged 75 years and above from April 2015 to March 2019. The national health insurance claims database, which covers most reimbursement claims in Japan, was selected as the primary data source. In this study, polypharmacy was defined as the simultaneous prescription of seven or more medications or multi-psychotropic medications. The primary outcome was the polypharmacy reduction ratio, which indicates the decrease in polypharmacy proportion based on the number of medications on an outpatient prescription. A total polypharmacy reduction of 19.3% for the “75–89 years” subgroup and 16.5% for the “90 years and above” subgroup was observed over four years. Based on prefecture analysis, the mean values of polypharmacy proportion showed a statistically significant reduction over four years. This study showed a successful nationwide reduction in polypharmacy prescriptions after implementing the polypharmacy management guidelines for older adults and incentive-based policies.

## 1. Introduction

Polypharmacy is a serious global health concern, primarily affecting older adults. It is associated with negative outcomes including adverse drug reactions, falls, and death [1]. Polypharmacy has been defined as the concurrent use of five or more medications in previous studies, and hyper-polypharmacy or excessive polypharmacy has been defined as the concurrent use of ten or more medications [2,3,4]. However, no standardized global definition of polypharmacy exists [5]. Japan has a rapidly aging population, and 33.5 million were aged 65 years or above among the total population of 127.1 million individuals in 2015 [6]. Polypharmacy is a serious issue among older adults in Japan [7]. In 2017, the Japan Gerontological Society and the Japan Geriatrics Society proposed a new definition of “older adults” as those aged 75 years and above based on the average life expectancy [8]. Previous Japanese research has argued that the traditional definition of older adults as those aged 65 years and above is no longer appropriate when evaluating polypharmacy in older adults. This is because the adverse events associated with polypharmacy primarily occur in adults aged over 80 years [9].

Recently, three important guidelines and incentive-based policies have been implemented to reduce polypharmacy among older adults. The Ministry of Health, Labour and Welfare (MHLW) published polypharmacy management guidelines for older adults in May 2018 titled “Proper Medication Guidelines for Older Adults” [10]. These guidelines included a flowchart of instructions for reviewing prescriptions with excessive medications and warnings about duplicate prescriptions and drug interactions, which is similar to the Clinical Medical Review (CMR) approach adopted in the Netherlands [11]. Furthermore, the MHLW periodically conducts medication fee amendments based on medication points with one point being equal to approximately 0.1 USD based on the currency ratio of 145.82 JPY/USD as of October 11th, 2022 [12]. The medication fee amendment of April 2016 included polypharmacy reduction incentives, such as 250 medication points (equivalent to approximately 17.1 USD) as a “comprehensive drug evaluation adjustment fee” for hospitals and clinics [13]. The medication point amendment of April 2018 added further polypharmacy reduction incentives, such as 125 medication points (equivalent to approximately 8.6 USD) as a “drug adjustment support fee” for pharmacies [14]. These incentives are awarded to clinics, hospitals, and pharmacies for each case where they were able to reduce two or more types of medications when a patient was previously receiving six or more medications in total [15]. Between April 2015 and March 2016 in Japan, the total national medical cost was 284.6 billion USD (equivalent to 41.5 trillion JPY), and the annual medication cost was approximately 61.0 billion USD (equivalent to 8.9 trillion JPY) [16]. Therefore, if these policies can successfully reduce polypharmacy, it would lead to a substantial economic impact.

We have previously reported on the effectiveness of the April 2016 medication point amendment, which led to a 7.3% reduction in polypharmacy nationwide [17]. A study indicated that the polypharmacy trend among older adults in Japan declined from 2013 to 2016, following an increasing trend from 2010 to 2013 [2]. Another study found that the 2018 amendment effectively reduced polypharmacy. However, the data for that study were drawn from a claims database maintained by a private company that does not include data from patients aged 75 years and above [18]. Therefore, information about recent long-term trends regarding nationwide polypharmacy in Japan that includes adults aged 75 years and above is lacking. We hypothesized that the polypharmacy management guidelines and incentive-based policies published by the MHLW in 2018 effectively reduced polypharmacy across Japan. We further posited that the impact of these guidelines and incentive-based policies is even stronger for older adults (75 years and above) than younger adults (under 75 years). This study aimed to confirm our hypotheses via long-term observations between April 2015 and March 2019 that focused on the older population. This study used the new definition of older adults as those aged 75 years and above provided by the Japan Gerontological Society and the Japan Geriatrics Society in 2017 [8].

## 2. Methods

### 2.1. Study Design and Primary Data Source

A serial cross-sectional study was conducted using open data from the National Database of Health Insurance Claims and Specific Health Checkups of Japan (NDB). The MHLW developed the NDB to collect data for health insurance claims from medical institutes in 2009. The NDB includes data on almost all patients, prescriptions, and hospitals in Japan, which are collected from medical reimbursement requests on a monthly basis [19]. The NDB has been used in several studies, but the MHLW places many restrictions on the NDB. Only those researchers who gain approval for their study protocols via a stringent expert review can access the NDB. Additionally, publication is mandatory to use the NDB, with penalties for commitment violation [20]. However, in October 2016, the MHLW revealed NDB Open Data without any restrictions, which provides aggregated information based on two types of age groups with a five-year range. It includes a sex (male or female) dataset and 47 prefectural datasets (sex and age information are not included for the prefectural datasets); however, detailed data, such as specific medications on a prescription and patient-level data, are not available [21].

### 2.2. Data Selection

We retrieved both five-year range age/sex data and prefecture data without age/sex information on the number of annual reimbursement claims for outpatient prescription fees from NDB Open Data [21]. The inpatient data were excluded from this study because of insufficient or missing data. The data of annual reimbursement claims for outpatient prescription fees were classified according to the period covered into Period 1 (April 2015–March 2016), Period 2 (April 2016–March 2017), Period 3 (April 2017–March 2018), and Period 4 (April 2018–March 2019), incorporating a total observation period of four years (April 2015–March 2019), as is shown in Figure 1. Moreover, Figure 1 presents the “Proper Medication Guideline for Older Adults” with instructions on reducing polypharmacy, and the medical service fee revisions, including incentives for polypharmacy reduction. Furthermore, Figure 1 shows the new definition of “older adults”, although it is not directly relevant to polypharmacy reduction.

### 2.3. Outcome Variable

In Japan, the basic outpatient prescription fee is 68 medication points for fewer than seven medications on a prescription, which is equivalent to approximately 4.7 USD. This excludes pro re nata medications administered for two weeks or less, which we defined as Category I for calculation purposes. To reduce polypharmacy, the MHLW created two outpatient prescription fee categories (Category II and Category III) in 2014 to denote an excess of medications on a prescription [22]. Category II is characterized by seven or more medications administered for longer than two weeks on a prescription; in this study, Category II was defined as polypharmacy. Multi-psychotropic drugs (three or more anxiolytics, three or more hypnotics, three or more antidepressants, three or more antipsychotics, or four or more anxiolytics/hypnotics as of April 2018) on a prescription are categorized as Category III; in this study, Category III was not recognized as polypharmacy to avoid the inclusion of prescriptions with fewer than five medications as polypharmacy.

Following the outcome measures used in our previous study [17], we calculated the polypharmacy proportion (PP) using Equation (1). Furthermore, we set the polypharmacy reduction ratio (PRR), calculated using Equation (2), as the outcome variable for this study (see below). All applicable raw data based on age and sex retrieved from NDB Open Data are summarized in Appendix A, and all applicable raw data based on prefectures retrieved from NDB Open Data are summarized in Appendix A.
(1)$$\mathrm{PP}=\frac{\mathrm{Annual}\;\mathrm{polypharmacy}\;\mathrm{prescriptions}\;\left(\mathrm{Category}\;\mathrm{II}\right)}{\mathrm{Annual}\;\mathrm{total}\;\mathrm{prescriptions}\;\left(\mathrm{CategoryI}+\mathrm{II}+\mathrm{III}\right)}$$
(2)$$\mathrm{PRR}=1-\frac{\mathrm{PP}\;\mathrm{after}\left(\mathrm{Period}\;2,\;\mathrm{Period}\;3,\;\mathrm{or}\;\mathrm{Period}\;4\right)}{\mathrm{PP}\;\mathrm{before}\left(\mathrm{Period}\;1,\;\mathrm{Period}\;2,\;\mathrm{or}\;\mathrm{Period}\;3\right)}$$

Figure 2 is an example of a case series from Period 1 to Period 4 that exhibits how each prescription is counted in NDB Open Data. A patient who received medications from four hospitals during Periods 1 to 4 is presented in Figure 2; the number of prescribed medicines administered for longer than two weeks is presented. Following the polypharmacy definition in this study, the polypharmacy prescriptions are highlighted in yellow in Figure 2. The red highlights in Figure 2 denote the patient’s annual number of polypharmacy prescriptions (Category II), which is 18 for Period 1, 13 for Period 2, 13 for Period 3, and 1 for Period 4 from among 21 total annual prescriptions.

### 2.4. Independent Variables

A multiple linear regression analysis considering each prefecture as an individual unit (*n* = 47) was conducted according to the analysis method used in our previous study [17]. The authors obtained the human variables of the population data and healthcare resource data for each prefecture disclosed in e-Stat, sponsored by the Ministry of Internal Affairs and Communications (MIC), and the number of medical facilities for each prefecture disclosed by the MHLW and MIC [23,24]. Five independent variables were set for the multiple regression analysis to identify the factors associated with the outcome variable. The proportion (%) of older adult residents (aged ≥ 65 years) was set as of 2015 based on data disclosed by the MIC because polypharmacy is usually a more serious issue in older populations, as is described in the MHLW guidelines [10]. Because sex is as fundamental a factor as age in epidemiology, the proportion (%) of male residents was set as of 2015 based on data disclosed by the MIC. The numbers of hospitals, clinics, and pharmacies per 100,000 residents were set as of 2015 based on data disclosed by the MHLW, as the incentives were introduced in 2016 and 2018 for these medical facilities.

### 2.5. Statistical Analysis

A serial cross-sectional analysis (PP and PRR trends with age and sex subgroups) was conducted using Microsoft Excel. For PRR, PRR1 shows the PP reduction ratio for Period 1–2, PRR2 for Period 2–3, PRR3 for Period 3–4, and PRR (long-term) for Period 1–4. Age subgroups were categorized as under 65 years, 65–74 years, 75–89 years, and 90 years and above to distinguish data consistent with the new definition of “older adults” in Japan [8]. Sex subgroups were categorized as male and female. 

Regarding the prefecture-based analysis (*n* = 47), the basic statistical information of the mean (95% confidence interval, hereafter 95% CI) of PP/PRR was described, and a paired t-test was conducted for the mean difference between PP (Period 1) and other PPs (Period 2, Period 3, and Period 4) by setting the mean PP (Period 1) as a reference with the TTEST procedure in SAS version 9.4. A box plot with the mean (95% CI) was described using JMP version 16.0. A colored map displaying the regional PP was also drawn according to each prefecture based on the NDB Open Data using a website-based free mapping tool (https://n.freemap.jp/, accessed on 27 July 2022) to aid the visualization and comprehension of the local impacts immediately after the implementation of the polypharmacy reduction policies and guidelines enforced in 2016 and 2018.

Additionally, a multiple linear regression analysis was performed to identify the factors among independent variables associated with long-term PRR using the REG procedure in SAS version 9.4 for Windows (SAS Institute, Cary, NC, USA). All *p* values were two-sided, and *p* < 0.05 was considered statistically significant.

### 2.6. Ethical Statement

In accordance with the current Ethical Guidelines for Medical and Health Research Involving Human Subjects in Japan and the ethical standards in the 1964 Declaration of Helsinki and its later amendments, an ethical committee review was not required for this study because NDB Open Data is publicly available and anonymously aggregated [25]. This study was conducted following the Strengthening the Reporting of Observational Studies in Epidemiology statement [26].

## 3. Results

### 3.1. PP Trends (April 2015–March 2019)

Demographic data, including populations for each subgroup and the number of annual reimbursement claims for outpatient prescription fees adjusted for the populations, are presented in Table 1. The PP trends for Periods 1–4 are summarized with age and sex subgroup analyses in Figure 3. For instance, in Period 1 (April 2015–March 2016), the PPs of the old (75–89 years) and super-old (90 years and above) subgroups were 7.8% and 9.7%, respectively, much higher than that of the total-population (4.1%). However, the male subgroup (4.4%), female subgroup (3.9%), and pre-old (65–74 years) subgroup (4.7%) did not show large deviations from the total-population (4.1%) in Period 1. Furthermore, the PP of the young (under 65 years) subgroup was 1.8%, which is much lower than that of the other subgroups. The PP trend among each subgroup was consistent throughout Periods 1–4, as is shown in Figure 3.

Figure 4 describes the PP in different prefectures in Period 1 (April 2015–March 2016) and the PP in different prefectures in Period 4 (April 2018–March 2019) to show the PP changes by prefecture over four years. We used red for ≥5.0%, yellow for 4.0–4.9%, and green for <4.0%. The red (≥5.0%) decreased from 9 to 0, and the green (<4.0%) increased from 15 to 38 after the policy implementation. Moreover, no prefectures worsened from yellow to red or from green to yellow during the period.

The prefecture-based PP trends over Periods 1 to 4 are summarized in Table 2. The mean values of PP are 4.40 for PP (Period 1), 4.01 for PP (Period 2), 4.02 for PP (Period 3), and 3.68 for PP (Period 4). Regarding the differences between the mean PPs and the reference PP (Period 1), all PPs (Period 2, Period 3, and Period 4) showed a statistically significant reduction, although PP (Period 2) and PP (Period 3) showed similar values.

### 3.2. PRR Trends (April 2015–March 2019)

The PRR trends, including the PRR (long-term), are summarized in Table 3. The total-population (all ages) demonstrated values of 8.3% in PRR1 and 8.4% in PRR3, which are much higher than 0.1% in PRR2. Furthermore, all subgroups consistently exhibited much lower values in PRR2 than in PRR1 and PRR3.

Regarding age subgroups, the old (75–89 years) subgroup reported values of 9.7% in PPR3 and 19.3% in PRR (long-term), and the super-old (90 years and above) subgroup exhibited values of 9.0% in PPR3 and 16.5% in PRR (long-term), which are higher than the values of 8.4% in PPR3 and 16.1% in PRR (long-term) for the total-population (all ages). The pre-old (65–74 years) subgroup reported values of 7.8% in PRR3 and 15.1% in PRR (long-term), which are lower than the values of 8.4% in PPR3 and 16.1% in PRR (long-term) for the total-population (all ages). The young (under 65 years) subgroup exhibited values of 7.7% in PRR3 and 15.3% in PRR (long-term), which are lower than the values of 8.4% in PPR3 and 16.1% in PRR (long-term) of the total-population (all ages).

Regarding sex, the female subgroup exhibited values of 9.6% in PRR3 and 18.5% in PRR (long-term), which are higher than the values of 7.2% in PPR3 and 13.5% in PRR (long-term) for male subgroups and values of 8.4% in PPR3 and 16.1% in PRR (long-term) for the total-population (all ages).

The prefecture-based PRR trends from Periods 1 to 4 with long-term PRR are summarized in Figure 5. The means of PRR1 and PRR3 were 8.75% and 8.42%, respectively, the mean of PRR2 was −0.30%, and the mean of long-term PRR was 16.18%. The highest prefectures for PRR1 (Fukui) and PRR3 (Ishikawa) had values of 11.8% and 12.3%, respectively, and the lowest prefectures for PRR1 (Okinawa) and PRR3 (Kagawa) had values of 5.8% and 3.9%, respectively. The highest prefecture in PRR2 (Yamanashi) had a value of approximately 2.2%, and the lowest prefecture for PRR2 (Ehime) had a value of −3.1%. The highest prefecture for long-term PRR (Yamanashi) showed a 21.1% polypharmacy reduction, and even the lowest prefecture in terms of long-term PRR (Kagawa) demonstrated an 8.3% polypharmacy reduction.

### 3.3. Associated Factor Analysis for Long-Term Polypharmacy Reduction Ratio over Four Years

Table 4 shows the results of the multiple linear regression analyses conducted to identify the factors associated with the effectiveness of the polypharmacy reduction policy. Long-term PRR was set as an outcome variable using the same independent variables as our previous study [17]. No factor showed a statistically significant association with long-term PRR.

## 4. Discussion

### 4.1. Key Findings

In this study, a statistically significant PP reduction was observed across four years after the release of the “Proper Medication Guideline for Older Adults” and the polypharmacy reduction incentives from the medical service fee revisions in 2016 and 2018, respectively. This study reveals more frequent polypharmacy in the “75 years and above” population in Japan, which is consistent with a previous report [2]. PRR3 shows a favorable reduction ratio of polypharmacy among all subgroups, and this is consistent with that observed in PRR1, as we previously reported [17]. This implies the effectiveness of incentives for pharmacies that were introduced in the medical fee amendment of 2018. A recent clinical trial in Japan reported the effectiveness of pharmacist interventions at reducing polypharmacy among older patients [27]. Moreover, the “75 years and above” population exhibited higher PRR3 than the total-population (all ages) and the “65–74 years” subgroup, further supporting the positive impact of the polypharmacy management guidelines for older adults [10].

As no reports providing recent long-term polypharmacy trends exist in Japan, this study identified PRR (long-term) trends among older adults. We found a successful reduction in polypharmacy among the “75 years and above” subgroups, with a more than 16% reduction in PRR (long-term). A similar study in New Zealand reported a 1–2% reduction in polypharmacy prevalence in older adults between 2014 and 2018 [28]. The female subgroup showed an approximately 5% greater reduction in PRR (long-term) than the male subgroup. These results are consistent with the higher repeated polypharmacy rate for men than women, as was reported in Taiwan [29]. Moreover, the “65–74 years” subgroup showed different trends from the “75 years and above” subgroups. Furthermore, they reported consistent or even lower long-term PRR than that for the total-population (all ages). Therefore, compared with the previous cut-off, the new cut-off that defines older adults as aged “75 years and above” accurately reflected individuals at higher risk for polypharmacy. The contents of the “Proper Medication Guidelines for Older Adults” in Japan are similar to the CMR approach to optimizing medication with the collaboration of pharmacists and physicians in the Netherlands [11]. By calculating the economic impact of this drastic reduction in polypharmacy, the direct cost for polypharmacy medications during Period 1 would be approximately 2.5 billion USD based on the assumption of 4.1% (PP in Period 1) of the total medication costs (61.0 billion USD) [4]. A nationwide 16.1% reduction in PRR (long-term) should have a substantial financial impact on the direct medication costs in Japan. The total cost benefit might be limited owing to the unmeasured labor costs of healthcare providers and incentives for medical facilities, as discussed in a previous cost-effectiveness study [30]. However, there are considerable clinical advantages to preventing drug-related problems due to polypharmacy or duplicated medications.

Regarding variances in PRR among prefectures, no factor demonstrated a statistically significant association with long-term PRR, and this result was partially different from the statistically significant association with PRR1 and other independent factors (proportion 65 years or above, number of hospitals per 100,000 residents, and number of clinics per 100,000 residents) exhibited in the results of our previous study [17]. This inconsistent finding might have been caused by the diversified characteristics of cities (i.e., big city, local city, or depopulating area) in the same prefecture. Therefore, it is necessary to confirm the polypharmacy trend in a more detailed sample such as secondary medical areas (SMAs) defined for regional emergency care by the MHLW, as SMA data have been available from the newly disclosed NDB Open Data (April 2019–March 2020) since August 2021 [21].

Finally, the results suggest that the “Proper Medication Guidelines for Older Adults” and the medical fee amendment of 2018 with incentives for medical facilities have effectively reduced polypharmacy in Japan. This reduction is particularly impressive when compared with gradually increasing polypharmacy trends in other countries reported by nationwide studies [4,29]. Additionally, we observed a successful long-term polypharmacy reduction for all prefectures (Figure 4). This implies that nationwide polypharmacy reduction policies or guidelines continuously had positive effects on all prefectures, and negative outcomes including adverse drug reactions, falls, and deaths associated with polypharmacy are expected to improve in Japan [1].

### 4.2. Strengths and Limitations

This study used valid and reliable real-world data unaffected by sampling biases to generate robust findings for polypharmacy trends in Japan. The NDB data incorporate more than 95% of medical insurance reimbursement claims in Japan [19]. While it is impossible to conduct a patient-based analysis in this study [19], NDB Open Data is quickly disclosed with a short lead time of approximately one to two years for data collection and cleaning by the MHLW, a much faster pace than that observed in recent studies from other countries [4,29]. Unfortunately, MHLW developed the system to monitor excessive medications based on medication reimbursement requests by each medical facility in Japan. Therefore, it is impossible to track patients who receive seven or more medications in total from multiple facilities using NDB Open Data if each prescription includes fewer than seven medications.

Additionally, patients’ background data other than age and sex, such as comorbidities, are unavailable in NDB Open Data. Although we discussed the financial impact of reducing polypharmacy above, quality of life data are lacking in NDB Open Data, meaning that we cannot conduct detailed economic analyses, such as a cost utility analysis using quality-adjusted life year. It is impossible to measure the change in adverse drug reactions, falls, and deaths associated with polypharmacy reduction through the current study’s design. Moreover, more detailed subgroups often show different characteristics from the results of ecological studies [31]. Thus, the polypharmacy trend analyzed using the prefecture level data might show different characteristics from more fragmented data such as secondary medical area data. However, NDB Open Data still provides a strong advantage over the original NDB, because under that system the MHLW only provided sampling datasets or limited regional data with an expert review of the study design and publication plan, taking over a year to confirm a data request for the original NDB [21]. Nevertheless, it is difficult to determine whether the polypharmacy management guidelines or the incentives provided by the 2018 medical fee amendment contributed more to polypharmacy reduction in older adults, even if we used the original NDB.

### 4.3. Future Perspectives

Further investigation using more smaller-segment data than prefecture data might provide good insights for a better understanding of the local polypharmacy situation in Japan. A recently published study on antibiotic use involved SMAs, but we did not find SMA-based analyses in the polypharmacy field [32]. Fortunately, the most recent NDB Open Data published in August 2021 includes SMA-level data [21]. Therefore, SMA-level analyses are now possible and would be helpful to develop local polypharmacy management strategies in Japan. Moreover, the magnitude of the impact of nationwide polypharmacy reduction on health outcomes must be investigated via carefully designed future studies.

## 5. Conclusions

This study revealed a successful reduction in nationwide polypharmacy across four years (April 2015–March 2019) among persons aged 75 years and above following the new definition of “older adults” in Japan after the “Proper Medication Guidelines for Older Adults” and the incentives for medical facilities were implemented in 2016 and 2018, respectively. A 19.3% reduction in polypharmacy was reported among persons aged 75–89 years, along with a 16.5% reduction in persons aged 90 years and above. Based on prefecture analysis, all prefectures exhibited obvious improvements in PP over four years (April 2015–March 2019). Further investigations should be conducted using more detailed ecological data, such as SMAs, to develop local polypharmacy management strategies in Japan.

## Figures and Tables

**Figure 1 ijerph-19-14684-f001:**
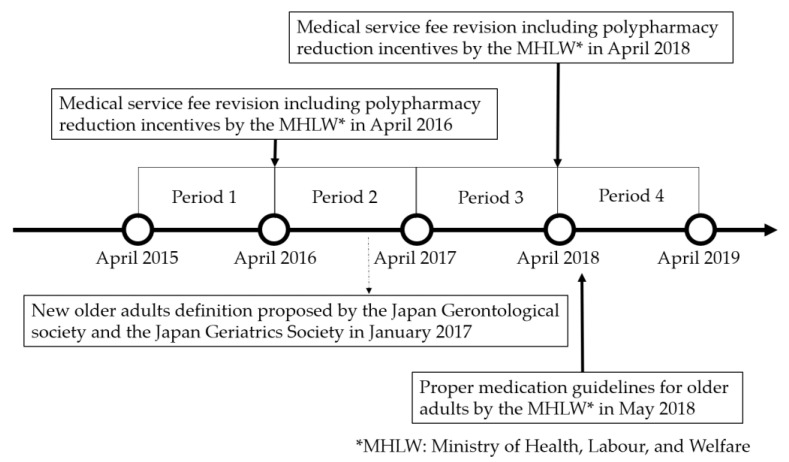
Research period timeline with relevant guidelines and policies.

**Figure 2 ijerph-19-14684-f002:**
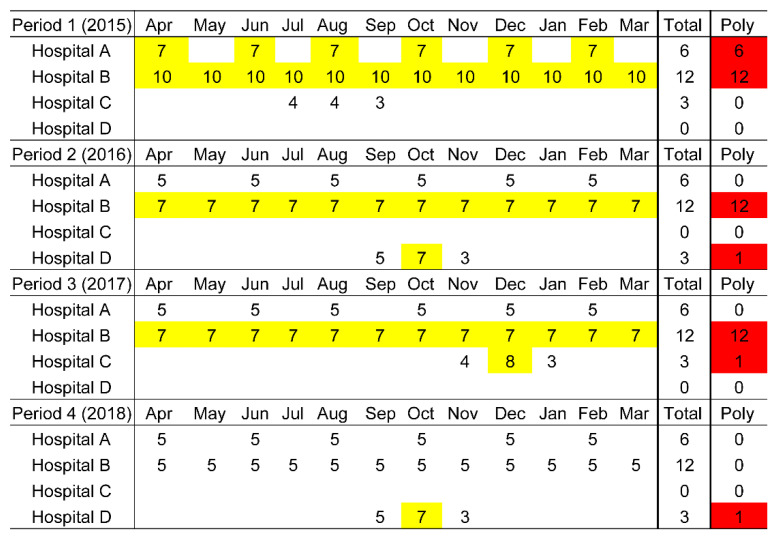
Case series example of total annual prescriptions and annual polypharmacy prescriptions during Periods 1 to 4.

**Figure 3 ijerph-19-14684-f003:**
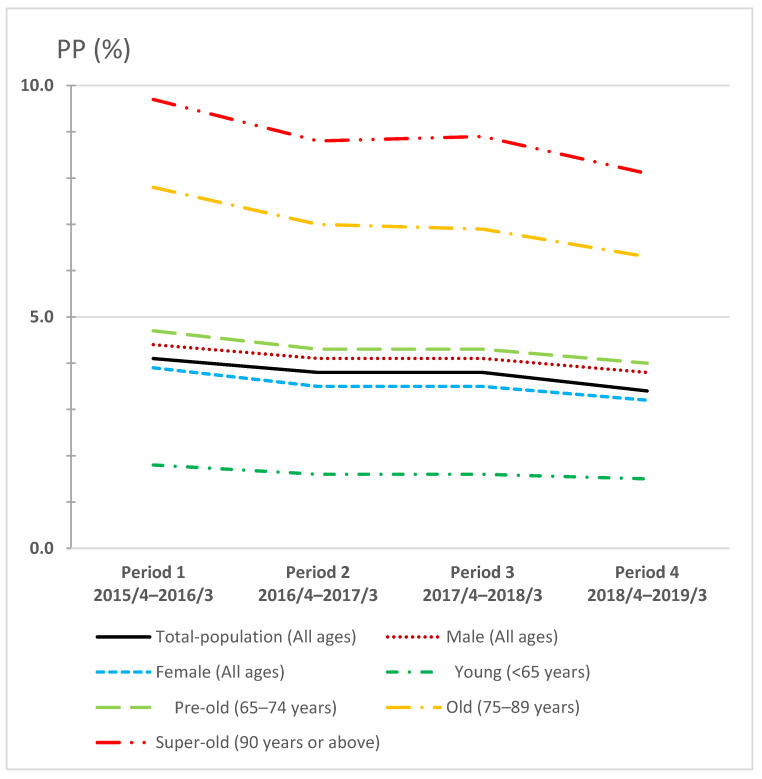
Age- and sex-based polypharmacy proportion trends across four years in Japan.

**Figure 4 ijerph-19-14684-f004:**
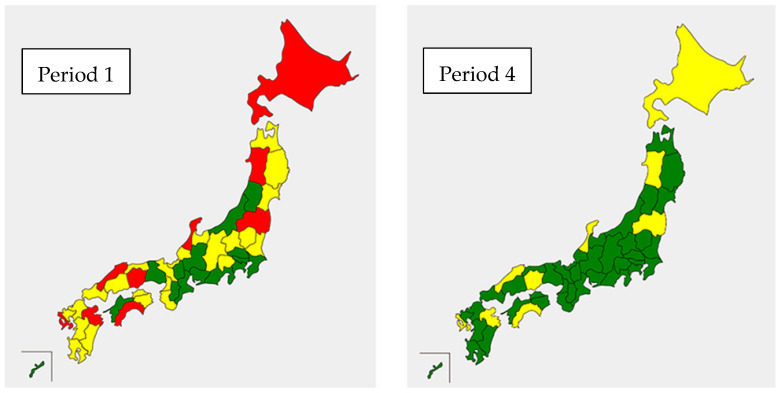
Prefecture-based polypharmacy proportion according to the third National Database of Health Insurance Claims and Specific Health Checkups of Japan Open Data for Period 1 (April 2015–March 2016) and Period 4 (April 2018–March 2019) with designated colors (red is used for 5.0% or above, yellow for 4.0–4.9%, and green for less than 4.0% polypharmacy proportion).

**Figure 5 ijerph-19-14684-f005:**
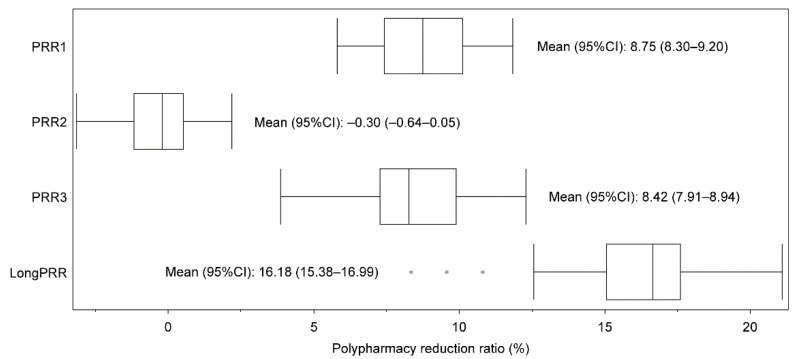
Prefecture-based box plot of polypharmacy reduction ratios with means.

**Table 1 ijerph-19-14684-t001:** Demographics of annual reimbursement claims and population over four years.

X §/Y ¶ (Population ‡)	Period 1 †	Period 2 †	Period 3 †	Period 4 †
Total-population (All ages)	246/5984 (126.8)	226/6019 (126.9)	229/6099 (126.7)	212/6148 (126.4)
Male (All ages)	235/5349 (61.7)	220/5393 (61.8)	225/5477 (61.7)	210/5522 (61.5)
Female (All ages)	255/6585 (65.2)	233/6613 (65.2)	234/6689 (65.0)	213/6741 (64.9)
Young (Under 65 years)	74/4182 (93.0)	68/4221 (92.4)	70/4298 (91.6)	65/4353 (90.9)
Pre-old (65–74 years)	418/8961 (17.5)	380/8801 (17.7)	377/8768 (17.7)	345/8704 (17.6)
Old (75–89 years)	1024/13,207 (14.5)	917/13,080 (15.0)	900/12,988 (15.4)	806/12,881 (15.8)
Super-old (90 years and above)	1139/11,718 (1.8)	1044/11,877 (1.9)	1046/11,736 (2.1)	939/11,565 (2.2)

Notes: † Period 1 (April 2015–March 2016), Period 2 (April 2016–March 2017), Period 3 (April 2017–March 2018), Period 4 (April 2018–March 2019). ‡ “Population” is presented in units of 1,000,000 residents, and data for each age subgroup are cited from the website (FY2015, FY2016, FY2017, and FY2018) of the Statistics Bureau of Japan. § X represents the amount of reimbursement claims applicable according to the polypharmacy definition in this study for outpatient prescription fees per 1000 residents. ¶ Y represents the amount of total reimbursement claims for outpatient prescription fees per 1000 residents. $\mathrm{PP}=\frac{\mathrm{Amount}\;\mathrm{of}\;\mathrm{annual}\;\mathrm{reimbursement}\;\mathrm{claims}\;\mathrm{in}\;\mathrm{category}\;\mathrm{II}}{\mathrm{Amount}\;\mathrm{of}\;\mathrm{annual}\;\mathrm{reimbursement}\;\mathrm{claims}\;\mathrm{in}\;\mathrm{categories}\;I+\mathrm{II}+\mathrm{III}}=\frac{X}{Y}$.

**Table 2 ijerph-19-14684-t002:** Prefecture-based polypharmacy proportion trends over 4 years.

	PP (Period 1)	PP (Period 2)	PP (Period 3)	PP (Period 4)
	2015/4–2016/3	2016/4–2017/3	2017/4–2018/3	2018/4–2019/3
Mean (95% CI) (%)	4.40 (4.21–4.59)	4.01 (3.84–4.18)	4.02 (3.85–4.18)	3.68 (3.53–3.83)
Difference (95% CI) (%)	Reference	−0.39 (−0.42–−0.36)	−0.38 (−0.42–−0.34)	−0.72 (−0.77–−0.66)
*p*-value	NA	<0.0001 *	<0.0001 *	<0.0001 *

Note: * *p* < 0.05 was considered statistically significant. PP (Period 1): polypharmacy proportion during April 2015–March 2016; PP (Period 2): polypharmacy proportion during April 2016–March 2017; PP (Period 3): polypharmacy proportion during April 2017–March 2018; PP (Period 4): polypharmacy proportion during April 2018–March 2019. $\mathrm{PP}=\frac{\mathrm{Amount}\;\mathrm{of}\;\mathrm{annual}\;\mathrm{reimbursement}\;\mathrm{claims}\;\mathrm{in}\;\mathrm{category}\;\mathrm{II}}{\mathrm{Amount}\;\mathrm{of}\;\mathrm{annual}\;\mathrm{reimbursement}\;\mathrm{claims}\;\mathrm{in}\;\mathrm{categories}\;I+\mathrm{II}+\mathrm{III}}=\frac{X}{Y}$.

**Table 3 ijerph-19-14684-t003:** Polypharmacy reduction ratio (PRR) over 4 years.

	PRR 1	PRR 2	PRR 3	PRR (Long Term)
Total-population (All ages)	8.3%	0.1%	8.4%	16.1%
Male (All ages)	7.3%	−0.5%	7.2%	13.5%
Female (All ages)	9.3%	0.7%	9.6%	18.5%
Young (less than 65 years old)	8.6%	−0.3%	7.7%	15.3%
Pre-old (65–74 years old)	7.4%	0.5%	7.8%	15.1%
Old (75–89 years old)	9.5%	1.2%	9.7%	19.3%
Super-old (90 years old or above)	9.5%	−1.4%	9.0%	16.5%

Note: $\mathrm{PRR}1=1-\frac{\mathrm{PP}\;\left(\mathrm{Period}\;2\right)}{\mathrm{PP}\;\left(\mathrm{Period}\;1\right)}$.
$\mathrm{PRR}2=1-\frac{\mathrm{PP}\;\left(\mathrm{Period}\;3\right)}{\mathrm{PP}\;\left(\mathrm{Period}\;2\right)}$.
$\mathrm{PRR}3=1-\frac{\mathrm{PP}\;\left(\mathrm{Period}\;4\right)}{\mathrm{PP}\;\left(\mathrm{Period}\;3\right)}$.
$\mathrm{PRR}\;\left(\mathrm{long}\;\mathrm{term}\right)=1-\frac{\mathrm{PP}\;\left(\mathrm{Period}\;4\right)}{\mathrm{PP}\;\left(\mathrm{Period}\;1\right)}$.

**Table 4 ijerph-19-14684-t004:** Associated factor analysis for long-term polypharmacy reduction ratio.

	Mean (95% CI)	Coefficient (95% CI)	*p*-Value
Long-term PRR	16.18 (15.38–16.99)		
X1 (2015)	28.31 (27.50–29.13)	0.31 (−0.66–1.29)	0.52
X2 (2015)	14.17 (13.57–14.76)	−0.09 (−1.48–1.31)	0.90
X3 (2015)	48.23 (47.94–48.52)	0.20 (−1.19–1.59)	0.78
X4 (2015)	47.33 (45.52–49.13)	0.03 (−0.14–0.20)	0.71
X5 (2015)	8.06 (7.11–9.02)	−0.07 (−0.46–0.31)	0.70

Note: X1: Proportion of older adults (65 years or above); X2: Proportion of male; X3: the number of pharmacies per 100,000 residents; X4: the number of hospitals per 100,000 residents; X5: the number of clinics per 100,000 residents; $\mathrm{Long}\;\mathrm{term}\;\mathrm{PRR}=1-\frac{\mathrm{PP}\;\left(\mathrm{Period}\;4\right)}{\mathrm{PP}\;\left(\mathrm{Period}\;1\right)}$ 95% CI: 95% confidence interval.

## Data Availability

As cited in [21], the primary data used are publicly disclosed by the Ministry of Health, Labour and Welfare.

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
