# Peer review of "Nationwide Long-Term Evaluation of Polypharmacy Reduction Policies Focusing on Older Adults in Japan"

_ijerph, 2022, doi:10.3390/ijerph192214684_

Round 1

Reviewer 1 Report (Previous Reviewer 1)

Thank you for addressing the comments.

Reviewer 2 Report (New Reviewer)

In my opinion the manuscript was corrected according to suggestions of Editor and can be published in present form.

This manuscript is a resubmission of an earlier submission. The following is a list of the peer review reports and author responses from that submission.

Round 1

Reviewer 1 Report

It is a well-described manuscript on a critical healthcare issue.

I suggest including future remarks/perspective will increase the impact of the manuscript and get better cited(recognized.

Reviewer 2 Report

As the deleterious consequences of senescence spare no one who reaches old age, the Reviewer finds the submitted work to be of significant interest to both medical professionals and lay persons alike.  Likewise, the potential and actual pitfalls associated with polypharmacy are well-reported across the Literature.  In the present paper, the Authors accurately and thoughtfully report a growing trend in curbing polypharmacy, in this instance, among Japan's aging population.  The statistical NBD data, as described, further validate the key point that a reduction in polypharmacy, in the studied population, appears to be underway.  Of interest as well, the Authors point out the closely-associated cost reduction incentives put in place by the nation's health insurance providers.  

Inasmuch as the subject matter deals with actual living beings, and not merely statistics, the Reviewer wonders as to the human consequence of this effort.   Likewise, one ponders the quality of life changes, survival statistics and other related metrics which, undoubtedly, must closely co-localize with significant modifications in presumably "necessary" medication. 

And it is here that the Reviewer must push back a bit against the Author's stated contention that NDB data fail to capture quality of life statistics.  Notwithstanding the limitations of NDB source material, it is axiomatic that, particularly in a highly-digitized civilization such as Japan, accurately captured sources for such information are readily available.  Furthermore, cross correlation between multiple data sources constitutes one of the most basic features of statistical analysis.

Concisely, in the Reviewer's viewpoint, the Author's failure to even lightly touch upon these crucial variables renders the manuscript divorced from the most important element in the equation - i.e. the consequence of monetarily incentivized polypharmacy reduction to the treated senescent patient.  Thus, the Reviewer believes the submitted work must address, not just the cost savings, but crucially, the human variable in the equation.  

Round 2

Reviewer 2 Report

It appears that either the Reviewer failed to clearly articulate their primary critical objections or the Authors failed to grasp the key message conveyed: "....[w]here in the manuscript is there any mention of the actual human consequence to the reduction in drugs described throughout the work in exquisite detail?".  Have the patients' lives been improved, unchanged, or degraded?  Was there an empirical status change to baseline lab values after the aforementioned (significant) reduction in drug intake?   

In the Reviewer's opinion - and after having the benefit of reading many published papers on the subject of polypharmacy, the lack of human context in this submission continues to constitute, in Reviewer's opinion,  an unacceptable blindspot.  Any interested Party reviewing the data is almost certainly going to wonder as to the human consequence of the changes wrought. 

Yet, such remains glaringly absent.  
